Barthelonids represent a deep-branching
metamonad clade with mitochondrion-related
organelles predicted to generate no ATP.
*Proc. R. Soc. B* **287**: 20201538.

evolution, genomics, taxonomy and
systematics

metamonada, phylogenomics,
mitochondrion-related organelles

**Authors for correspondence:**
Euki Yazaki
e-mail: euki87@gmail.com
Yuji Inagaki
e-mail: yuji@ccs.tsukuba.ac.jp

Electronic supplementary material is available
online at https://doi.org/10.6084/m9.figshare.
c.5097098.

# Barthelonids represent a deep-branching metamonad clade with mitochondrion-related organelles predicted to generate no ATP

Euki Yazaki[1], Keitaro Kume[2], Takashi Shiratori[3,4], Yana Eglit[6,7], Goro Tanifuji[8],
Ryo Harada[4], Alastair G. B. Simpson[6,7], Ken-ichiro Ishida[3,4],
Tetsuo Hashimoto[3,4] and Yuji Inagaki[4,5]

[1]Interdisciplinary Theoretical and Mathematical Sciences (iTHEMS), RIKEN, Wako, Saitama, Japan
[2]Faculty of Medicine, [3]Faculty of Life and Environmental Sciences, [4]Graduate School of Life and Environmental
Sciences, and [5]Center for Computational Sciences, University of Tsukuba, Tsukuba, Ibaraki, Japan
[6]Department of Biology, and [7]Centre for Comparative Genomics and Evolutionary Bioinformatics,
Dalhousie University, Halifax, Nova Scotia, Canada
[8]Department of Zoology, National Museum of Nature and Science, Ibaraki, Japan

EY, 0000-0003-4962-3340; AGBS, 0000-0002-4133-1709; YI, 0000-0003-0101-8483

We here report the phylogenetic position of barthelonids, small anaerobic
flagellates previously examined using light microscopy alone. *Barthelona*
spp. were isolated from geographically distinct regions and we established
five laboratory strains. Transcriptomic data generated from one *Barthelona*
strain (PAP020) were used for large-scale, multi-gene phylogenetic (phylo-
genomic) analyses. Our analyses robustly placed strain PAP020 at the base
of the Fornicata clade, indicating that barthelonids represent a deep-branching
metamonad clade. Considering the anaerobic/microaerophilic nature of
barthelonids and preliminary electron microscopy observations on strain
PAP020, we suspected that barthelonids possess functionally and structurally
reduced mitochondria (i.e. mitochondrion-related organelles or MROs). The
metabolic pathways localized in the MRO of strain PAP020 were predicted
based on its transcriptomic data and compared with those in the MROs of
fornicates. We here propose that strain PAP020 is incapable of generating
ATP in the MRO, as no mitochondrial/MRO enzymes involved in substrate-
level phosphorylation were detected. Instead, we detected a putative
cytosolic ATP-generating enzyme (acetyl-CoA synthetase), suggesting that
strain PAP020 depends on ATP generated in the cytosol. We propose two
separate losses of substrate-level phosphorylation from the MRO in the
clade containing barthelonids and (other) fornicates.

## 1. Introduction

Elucidating the evolutionary relationships among the major groups of eukaryotes is
one of the most fundamental but unsettled questions in biology. It is widely accepted
that large-scale molecular data for phylogenetic analyses (so-called phylogenomic
data) are indispensable to infer ancient splits in the tree of eukaryotes [1–3]. Preparing
phylogenomic data has been greatly advanced by the recent technological improve-
ments in sequencing that generate a large amount of molecular data at an affordable
cost and in a reasonable time-frame [4,5]. Further, some recent phylogenomic ana-
lyses have included uncultured microbial eukaryotes (e.g. [6]), since the libraries
for sequencing of the whole-genome/transcriptome can be prepared from a small
number of cells (or even a single cell) isolated from an environmental sample [7,8].

Despite these advances in experimental techniques, it is realistic to assume that
no current phylogenomic analysis has covered the true diversity of eukaryotes.

**Figure 1.** Light micrographs of *Barthelona* spp. examined in this study. Strains PAP020, FB11, LRM2, EYP1702 and PCE are shown in (*a*–*e*), respectively. Flagella are marked by arrowheads. Scale bars, 10 μm.

A large number of extant microbial eukaryotes have never been examined using transcriptomic or genomic techniques, and some of them may hold the keys to resolving important unanswered questions in eukaryotic phylogeny and evolution. Thus, to reconstruct the evolutionary relationships among the major eukaryotic assemblages to a resolution that is both accurate and informative, the taxon sampling in phylogenomic analyses has been improved by targeting two classes of organisms: (i) novel microbial eukaryotes that represent lineages that were previously unknown to science, and (ii) 'orphan eukaryotes' that had been reported before, but whose evolutionary affiliations were unresolved by morphological examinations and/or single-gene phylogenies [6,8–15].

Many of these 'orphan eukaryotes' were described based solely on morphological information prior to the regular use of gene sequences in phylogenetic/taxonomic studies. One such organism is the small free-living heterotrophic biflagellate *Barthelona vulgaris* [16]. The initial description of *B. vulgaris* was based on light microscopy observations of cells isolated from marine sediment from Quibray Bay, Australia, and maintained temporarily in nominally anoxic crude culture [16]. The morphospecies was later identified at different geographical locations [17,18] but never examined with methods incorporating molecular data. These past studies identified no special morphological similarity between *B. vulgaris* and any eukaryotes described to date [16–18]. Thus, to clarify the phylogenetic placement of *Barthelona* in the tree of eukaryotes, molecular phylogenetic analyses are required, preferably at the 'phylogenomic' scale.

We here report five laboratory strains of *Barthelona* (EYP1702, FB11, LRM2, PAP020 and PCE; figure 1*a*–*e*) isolated from separate geographical regions, and infer their phylogenetic positions assessed by analysing both small subunit ribosomal DNA (SSU rDNA) and phylogenomic data. A SSU rDNA phylogeny robustly united all of the *Barthelona* strains together, but the precise placement of *Barthelona* spp. among other eukaryotes remained inconclusive. To infer the precise phylogenetic position of barthelonids, we obtained a transcriptome data from strain PAP020, and analysed its phylogenetic position from a eukaryote-wide dataset containing 148 genes. The transcriptome data of strain PAP020 were also used for reconstructing the metabolic pathways in a functionally and structurally reduced mitochondrion that is the result of adaptation to anaerobiosis.

## 2. Material and methods

### (a) Isolation and cultivation

We established five laboratory strains of *Barthelona* sp. in this study (figure 1*a*–*e*). Strains PAP020 and EYP1702 (figure 1*a*,*d*) were isolated from anaerobic mangrove sediments collected at a seawater lake in the Republic of Palau in November 2011 and October 2017, respectively. The laboratory cultures have been maintained in mTYGM-9 medium (http://mcc.nies.go.jp/medium/ja/mtygm9.pdf) with prey bacteria at 18–20°C. An anaerobic environment within the laboratory cultures was created by the respiration of prey bacteria. LRM2 (figure 1*b*) was isolated from mud of a defunct saltern (now normal salinity) on the Ebro Delta near San Carles de la Ràpita, Catalonia, Spain, in February 2015. FB11 (figure 1*c*) was isolated from False Bay, an intertidal mud flat on San Juan Island, WA, USA, in June 2015. PCE (figure 1*e*) was isolated from intertidal sediment near Cavendish, PEI, Canada, in July 2016. The established cultures were maintained with co-cultured bacteria on 3% LB in sterile natural seawater at 18–21°C.

### (b) SSU rDNA phylogenetic analysis

Total DNA samples of *Barthelona* sp. strains PAP020, EYP1702, FB11, PCE and LRM2 were extracted from the cultured cells using a DNeasy Plant mini kit (Qiagen) or NucleoSpin Tissue kit (Macherey-Nagel). Near-complete SSU rDNA fragments were amplified from each DNA sample by PCR, using either primers SR1 and SR12 [19] or 18F and 18R [20]. The amplification programme consisted of 30 cycles of denaturation at 94°C for 30 s, annealing at 55°C for 30 s and extension at 72°C for 90 s. The amplified product was gel-purified, cloned and sequenced by the Sanger method.

We aligned the SSU rDNA sequences of the five *Barthelona* strains with those of 91 phylogenetically diverse eukaryotes by using MAFFT v. 7.205 [21,22]. After manual exclusion of ambiguously aligned positions, 1573 nucleotide positions were subjected to maximum-likelihood (ML) phylogenetic analyses by using IQTREE v. 1.5.4 [23] with the GTR + R6 model, with ML bootstrap percentage values (MLBPs) derived from 500 non-parametric bootstrap replicates. The SSU rDNA alignment was also subjected to Bayesian phylogenetic analysis using MrBayes v. 3.2.3 [24] with GTR + Γ model. The Markov chain Monte Carlo (MCMC) run was performed with one cold and three heated chains with default chain temperatures. We ran 3 000 000 generations, and sampled log-likelihood scores and trees with branch lengths every 1000 generations (stationarity was confirmed by plotting the log-likelihoods sampled during the MCMC). The first 25% generations were discarded as burn-in. The consensus tree with branch lengths and BPPs were calculated from the remaining trees.

### (c) RNA-seq analyses

We conducted two RNA-seq runs of *Barthelona* sp. strain PAP020. The sequence reads from the first analysis were used for a phylogenomic analysis assessing the position of *Barthelona* spp. in the tree of eukaryotes, while those from the second sequencing run were used for surveying the proteins localized in the mitochondrion-related organelle (MRO) in strain PAP020 (see below). As bacteria in the culture medium of strain PAP020 were eliminated carefully before the RNA preparation for the second RNA-seq

analysis (see below), we anticipated that the second transcriptome data would much less contaminated by bacterial sequences than the first one, and thus be more suitable for predicting the metabolic pathways in the MRO.

For the first RNA-seq run, PAP020 cells, together with bacterial cells in the culture medium, were harvested and subjected to RNA extraction using TRIzol (Life Technologies) by following the manufacturer's protocol. We shipped the RNA sample to a biotech company (Hokkaido System Science) for cDNA library construction from the poly-A tailed RNAs and subsequent sequencing using the Illumina HiSeq 2500 platform, which generated $2.9 \times 10^7$ paired-end 100 bp reads (2.9 Gb in total). The initial reads were then assembled into 29 251 unique contigs by TRINITY [25,26].

For the second RNA-seq run, we separated PAP020 cells from the bacterial cells in the culture medium by a gradient centrifugation using Optiprep (Axis Shield), as reported previously [27], with slight modifications (the Optiprep solution containing the eukaryotic cells and bacteria was centrifuged at $2000g$ for 20 min, instead of $800g$ for 20 min). Total RNA was extracted from the harvested eukaryote-enriched fraction, using TRIzol, by following the manufacturer's protocol. Poly-A tailed RNAs in the RNA sample described above were purified with a Dynabeads mRNA Purification Kit (Thermo Fisher Scientific), and then used to construct the cDNA library using the SMART-Seq v4 Ultra Low Input RNA Kit for Sequencing (Takara Bio USA) and Nextera XT DNA Library Preparation Kit (Illumina). The resultant cDNA library was sequenced with the Illumina Miseq platform, yielding $3.7 \times 10^7$ paired-end 300 bp sequence reads (8.6 Gb in total). These were assembled into 21 286 unique contigs using TRINITY.

## (d) Phylogenomic analyses

To elucidate the phylogenetic position of *Barthelona* sp. strain PAP020, we prepared a phylogenomic alignment by updating an existing dataset comprising 157 genes (see electronic supplementary material, table S1) [10,15,28]. For each of these 157 genes, we added the homologous sequences retrieved from the transcriptomic data of strain PAP020 (this study) and four fornicates (*Carpediemonas membranifera*, *Aduncisulcus paluster*, *Kipferlia bialata* and *Dysnectes brevis*; [29]). Each single-gene alignment was aligned individually by MAFFT v. 7.205 with the L-INS-i algorithm followed by manual correction and exclusion of ambiguously aligned positions. For each of the single-gene alignments, the ML phylogenetic tree was inferred by RAxML v. 8.1.20 [30] under the LG + Γ + F model with robustness assessed with a 100 replicate bootstrap analysis.

Individual single-gene trees were inspected to identify the alignments bearing aberrant phylogenetic signal that disagreed strongly with any of a set of well-established monophyletic assemblages in the tree of eukaryotes, namely Opisthokonta, Amoebozoa, Alveolata, Stramenopiles, Rhizaria, Rhodophyta, Chloroplastida, Glaucophyta, Haptophyta, Cryptophyta, Jakobida, Euglenozoa, Heterolobosea, Diplomonadida, Parabasalia and Malawimonadidae. Nine out of the 157 single-gene alignments were found to bear idiosyncratic phylogenetic signal and were excluded from the phylogenomic analyses described below. After inspection of single-gene alignments/trees, the remaining 148 single-gene alignments (electronic supplementary material, table S1) were concatenated into a single phylogenomic alignment containing 83 taxa with 38 816 unambiguously aligned amino acid positions (148-gene alignment). The coverage for each single-gene alignment is summarized in electronic supplementary material, table S1.

ML analyses of 148-gene alignment were conducted by using IQTREE v. 1.5.4 with the LG + Γ + F + C60 + PMSF (posterior mean site frequencies) model [31] and robustness evaluated with an ML bootstrap analysis on 100 replicates. We also conducted a Bayesian phylogenetic analysis with the CAT + GTR model using PHYLO-BAYES v. 1.5a [32–34]. In this analysis, two MCMC runs were run

for 5000 cycles with 'burn-in' of 1250 ('maxdiff' value was 0.96743). The consensus tree with branch lengths and Bayesian posterior probabilities (BPPs) were calculated from the remaining trees.

The phylogenetic position of *Barthelona* sp. strain PAP020 inferred from the 148-gene alignment was assessed by an approximately unbiased (AU) test [35]. We modified the ML tree to prepare four alternative tree topologies, in which strain PAP020 branches (i) at the base of the Parabasalia clade, (ii) at the base of the clade of parabasalids and fornicates, (iii) with *Paratrimastix pyriformis*, and (iv) at the base of the Metamonada clade. Site likelihood data were calculated over each of the five trees examined (ML plus four alternative trees) using IQTREE and then analysed in CONSEL v. 0.20 [36] with the default settings.

## (e) Fast-site removal and gene subsampling analyses

We evaluated the contribution of fast-evolving sites in the 148-gene alignment to the position of *Barthelona* sp. strain PAP020. Individual rates for sites were calculated over the ML tree topology using DIST_EST [37] with the LG + Γ + F model. Fast-evolving sites were progressively removed from the original 148-gene alignment in 4000-position increments, and each of the resulting alignments was subjected to 100 replicate rapid ML bootstrap analysis with RAxML v. 8.1.20 with the LG + Γ + F model.

To evaluate the impact of gene subsampling on the position of strain PAP020 deduced from the 148-gene alignment, we conducted the analyses described below. Thirty out of the 148 genes were randomly sampled and concatenated into a single alignment. A '30-gene' alignment was then subjected to an ML bootstrap analysis (using the UFBOOT approximation with 1000 replicates) using IQTREE with the LG + F + Γ model. This procedure was repeated 60 times. Similarly, we subsampled 60 genes for 30 times, 90 genes for 20 times and 120 genes for 10 times, and all of these alignments were subjected to the ML bootstrap analysis described above.

## (f) Transmission electron microscopy observation of *Barthelona* sp. strain PAP020 cells

We conducted preliminary transmission electron microscopy (TEM) observation of *Barthelona* sp. strain PAP020, focusing on the MROs. Cultivated cells were centrifuged and fixed for 1 h at room temperature with a mixture of 2% (v/v) glutaraldehyde, 0.1 M sucrose and 0.1 M sodium cacodylate buffer (pH 7.2, SCB). Fixed cells were washed with 0.2 M SCB three times. Cells were post-fixed with 1% (v/v) $OsO_4$ with 0.1 M SCB for 1 h at 4°C, then washed with 0.2 M SCB two times. Dehydration was performed using a graded series of 30–100% ethanol (v/v). After dehydration, cells were placed in a 1:1 mixture of 100% ethanol and acetone for 10 min and acetone for 10 min for two cycles. Resin replacement was performed by a 1:1 mixture of acetone and Agar Low Viscosity Resin R1078 (Agar Scientific Ltd, Stansted, UK) for 30 min and resin for 2 h. Resin was polymerized by heating at 60°C for 8 h. Ultrathin sections were prepared on a Reichert Ultracut S ultramicrotome (Leica, Wetzlar, Germany), double stained with 2% (w/v) uranyl acetate and lead citrate [38,39], and observed using a Hitachi H-7650 electron microscope (Hitachi High-Technologies Corp., Tokyo, Japan) equipped with a Veleta TEM CCD camera (Olympus Soft Imaging System, Münster, Germany).

## (g) Prediction of proteins localized in the mitochondrion-related organelle in *Barthelona* sp. strain PAP020

We searched for mRNA sequences encoding proteins predicted to be localized to the MRO in *Barthelona* sp. strain PAP020, as

well as those involved in anaerobic ATP generation. For this, we searched among the contigs generated from the second RNA-seq experiment by TBLASTN, using the hydrogenosomal/MRO proteins in *Trichomonas vaginalis*, *Giardia intestinalis* and other fornicates (e.g. *K. bialata* and *D. brevis*) [29], as well as the mitochondrial proteins in *Saccharomyces cerevisiae* [40], as the queries. The amino acid sequences deduced from the contigs retrieved by the first BLAST searches were then subjected to BLASTP analyses against the NCBI nr database to exclude false positives. The domain structures of the putative MRO proteins were examined using hmmscan v. 3.1 (http://hmmer.org). We inspected each of the putative MRO proteins for potential mitochondrial targeting sequences using MitoFates [41] with default parameters for the fungal sequences, and NommPred [42] with parameters for canonical mitochondria and MRO. The transcript levels of MRO gene candidates were calculated for transcripts per kilobase million (TPM) by RSEM [43].

# 3. Results and discussion

## (a) Phylogenetic position of barthelonids

SSU rDNA sequences are valuable phylogenetic markers for elucidating the close relatives of a eukaryote of interest, but do not reliably resolve all deeper splits in the eukaryotic phylogeny. Overall, the ML and Bayesian phylogenetic analyses of SSU rDNA resolved known major eukaryote groups with moderate to high statistical support values. However, the SSU rDNA phylogeny failed to recover the monophyly of Amoebozoa or Apusomonadida, probably because long-branch sequences (e.g. *Dictyostelium discoideum* and *Thecamonas trahens*) were placed unstably, and in aberrant positions on the ML tree. Deeper-order phylogenetic relationships, such as Amorphea, SAR, Cryptista, Haptista, Discoba, Metamonada and CRuMs, most of which were reconstructed exclusively in phylogenomic studies, were also not recovered in the SSU rDNA analysis shown here (figure 2), due to insufficient phylogenetic signal in the single-gene (SSU rDNA) alignment.

In the SSU rDNA tree, all of the *Barthelona* sp. strains (PAP020, EYP1702, FB11, PCE and LRM2) grouped together with an ML bootstrap value (MLBP) of 83% and a BPP of 0.98. In this *Barthelona* clade, strains EYP1702 and PCE were the earliest and second earliest diverging taxa, respectively, and strains PAP020, LRM2 and FB11 formed a tight subclade. The *Barthelona* clade was sister to a Fornicata clade comprising *C. membranifera*, *K. bialata*, *D. brevis*, *Retortamonas* sp. and *Giardia intestinalis* (figure 2), but statistical support was equivocal (MLBP 56%; BPP 0.86). This possible affinity between *Barthelona* and fornicates in the SSU rDNA phylogeny is provocative, as both lineages thrive in oxygen-poor environments and possess double-membrane-bounded MROs instead of typical mitochondria (see figure 4a for the putative MRO in strain PAP020) [43–48]. Thus, we took a phylogenomic approach to resolve the position of barthelonids more robustly within the tree of eukaryotes.

As anticipated, both ML and Bayesian phylogenetic analyses of a multi-gene alignment comprising 148 genes (148-gene alignment) provided deeper insights into the backbone of the tree of eukaryotes (figure 3a) than the SSU rDNA analyses (figure 2). The backbone tree topology and statistical support values (figure 3a) agreed largely with those reported in prior studies [10,15,28], which analysed multi-gene alignments generated from the same core set of 157 single-gene alignments with mostly similar taxon sampling. The topology

includes well-established clades including SAR, Amorphea, Cryptista and Discoba, but, as is common, did not infer a monophyletic Archaeplastida [8,49]. Likewise, the 148-gene phylogeny recovered neither the clade of *Telonema subtilis* and SAR (T-SAR) [8] nor that of centrohelids and haptophytes (Haptista) [11]. We suspect that large proportions of missing data in the sequence of *T. subtilis* and the single included centrohelid (66% and 65% missing data, respectively), which derived from the transcriptomic data generated by 454 pyrosequencing [50], hindered the recoveries of T-SAR and Haptista in the 148-gene phylogeny.

The 148-gene phylogeny grouped *Barthelona* sp. strain PAP020 and 6 fornicates together with an MLBP of 99% and a BPP of 1.0 (figure 3a). In this clade, strain PAP020 occupied the basal position, which was supported fully by both ML and Bayesian analyses. The clade of strain PAP020 and fornicates was connected sequentially with parabasalids (MLBP 100%; BPP 0.70), then with *Paratrimastix pyriformis* (representing Preaxostyla), to form the Metamonada clade with an MLBP of 98% and a BPP of 0.98 (figure 3a). Support for these relationships was hardly affected by exclusion of rapidly evolving alignment positions, until greater than 60% of sites were excluded (electronic supplementary material, figure S2). We also performed gene subsampling analyses to evaluate the potential heterogeneity of phylogenetic signals among the genes [51]. In the analyses of the alignments comprising 30 randomly sampled genes (electronic supplementary material, figure S3A), strain PAP020 displayed the phylogenetic affinity to either parabasalids or fornicates, implying that two conflicting phylogenetic signals, one uniting strain PAP020 and parabasalids and the other uniting strain PAP020 and fornicates, are present in the genes considered here. We checked the position of strain PAP020 in the 30-gene analyses in which neither the affinity of strain PAP020 to fornicate nor parabasalids received MLBPs greater than 50% (see the data points highlighted by red arrowheads in electronic supplementary material, figure S3A). The ML analyses of five out of the seven alignments corresponding to the aforementioned data points recovered the clade of strain PAP020, fornicates, and the relatively long-branching, data-poor rhizarian *Quinqueloculina* sp., which was obviously misplaced, with high statistical support (electronic supplementary material, figure S4A–G). We suspected that *Quinqueloculina* sp. was artefactually attracted to fornicates (and strain PAP020) by long-branch attraction (LBA) [52,53]. Significantly, the putative LBA artefact appeared to be overcome and PAP020 branching with fornicates increasingly dominated as the number of sampled genes was increased to 60 then 90, while all samples of 120 genes except one recovered a PAP020 + Fornicata clade with high statistical support (electronic supplementary material, figure S3B–D). Altogether, we could find little evidence to regard the phylogenetic affinity between strain PAP020 and fornicates as an LBA artefact.

We applied the AU test to the ML tree and four alternative trees, wherein strain PAP020 branched at the base of (i) the Parabasalia clade, (ii) the clade of Fornicata + Parabasalia, (iii) the Metamonada clade, and (iv) Preaxostyla (i.e. PAP020 was sister to *Paratrimastix*), and all of the alternative trees were rejected ($p = 0.0000$; see the inset of figure 3b). In summary, the results from the phylogenetic analyses of the 148-gene alignment consistently and robustly indicated that barthelonids are a previously overlooked Metamonada lineage, which has a specific affinity with the Fornicata clade.

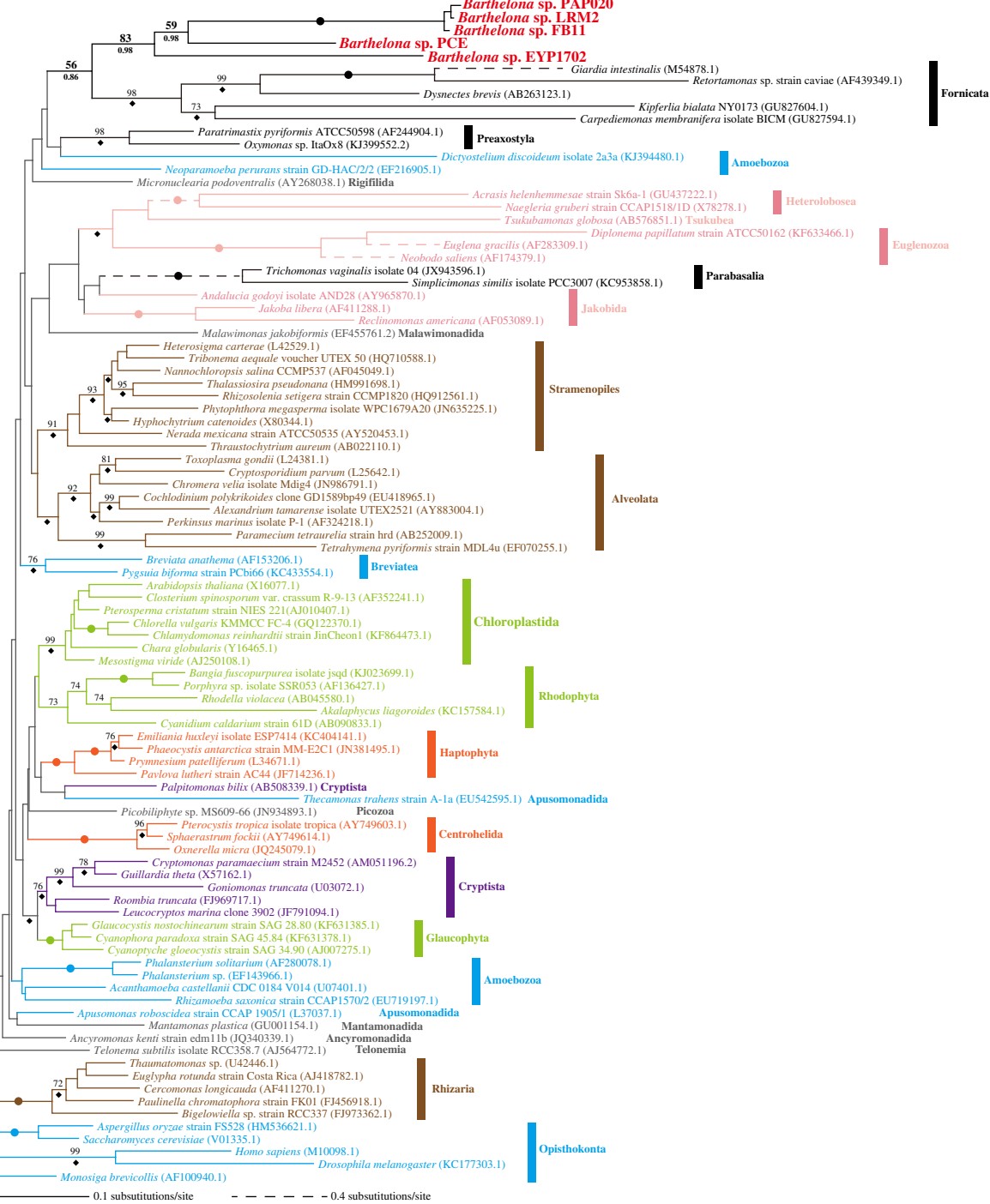

**Figure 2.** Global eukaryotic phylogeny inferred from small subunit ribosomal DNA sequences. The tree topology was inferred using the ML method and MLBPs and BPPs were mapped on the ML tree. The nodes marked by dots were supported by MLBPs of 100% and BPPs of 1.0. MLBPs less than 70% are not shown unless related to the position of *Barthelona* spp. BPPs of 0.95 or more are marked by diamonds. (Online version in colour.)

## (b) No ATP production in the mitochondrion-related organelle of *Barthelona* sp. PAP020

All of the *Barthelona* strains assessed in this study (strains PAP020, EYP1702, PCE, LRM2 and FB11) are grown under oxygen-poor conditions in the laboratory. Our preliminary ultrastructural observation of strain PAP020 did not reveal a typical mitochondrion. Instead, we observed a densely stained, double membrane-bounded organelle (figure 4a). As all meta-monads studied so far lack typical mitochondria, we suspect that the double membrane-bounded organelle identified in strain PAP020 is the MRO. Consistent with the anaerobic/microaerophilic characteristics of the barthelonid strains, the BLAST search considering yeast mitochondrial proteins as

queries detected almost no transcripts encoding any proteins comprising the electron transfer chain or ATP synthase, which are required for ATP generation under aerobic conditions, in the transcriptome data of strain PAP020, except NADH-quinone oxidoreductase subunits E and F (NuoE and F; figure 4b).

According to the phylogenetic position of barthelonids deduced from the SSU rDNA and 148-gene phylogeny (figures 2 and 3), the metabolic pathways retained in the barthelonid MROs are significant to infer the evolutionary history of the MROs in the Fornicata clade. Leger *et al.* [29] proposed that the ancestral fornicate species possessed an MRO with a metabolic capacity similar to that of the hydrogenosomes in parabasalids like *Trichomonas vaginalis* (The metabolic

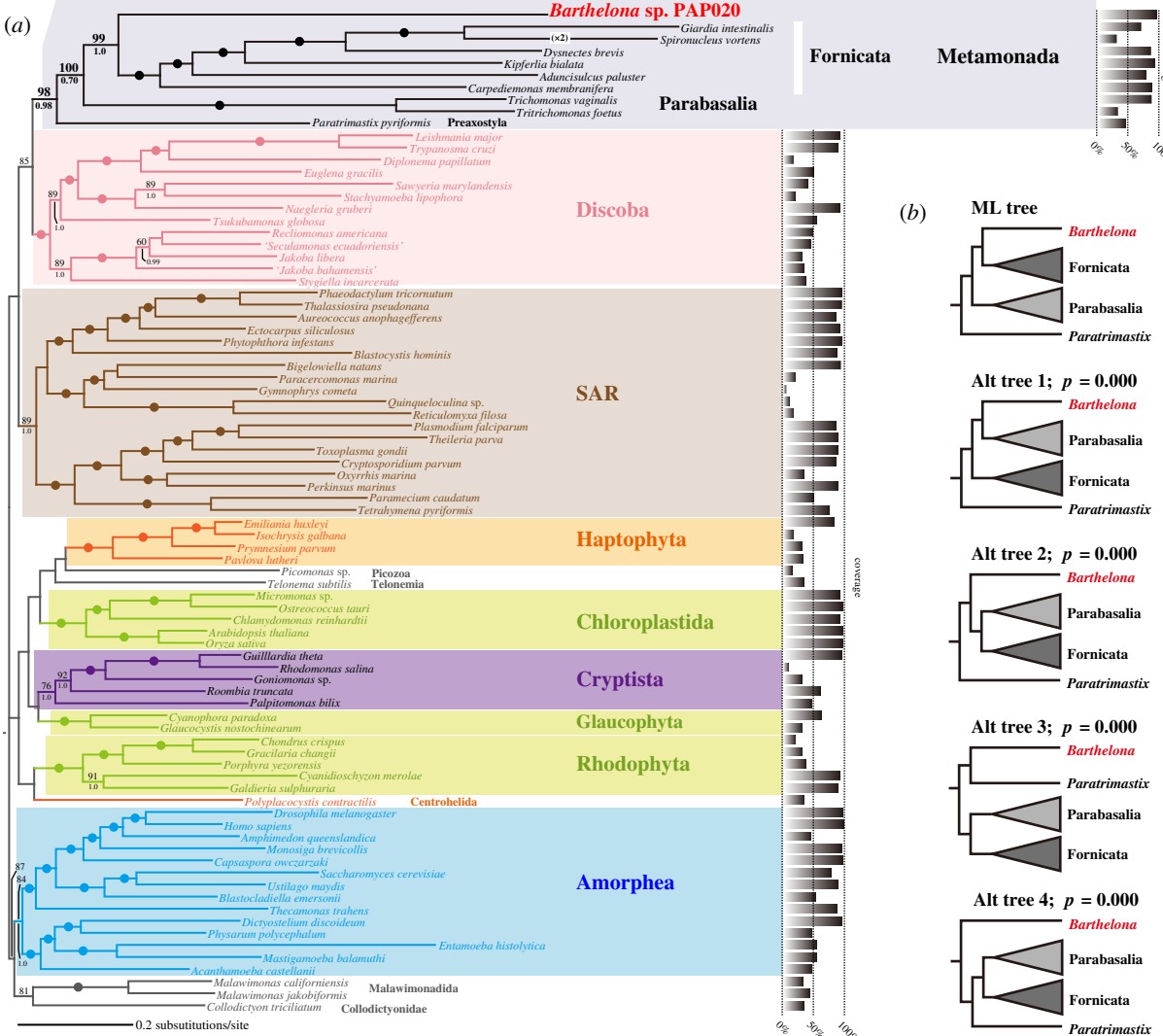

**Figure 3.** Global eukaryotic phylogeny inferred from a 148-gene alignment. (*a*) The tree topology inferred using the ML method. MLBPs and BPPs were mapped on the ML tree. The Bayesian analysis recovered an identical overall topology. The nodes marked by dots were supported by MLBPs of 98% or more, and BPPs of 0.95 or more. MLBPs less than 60% and BPPs below 0.80 are not shown outside Metamonada. The bar graph for each taxon indicates the per cent coverage of the amino acid positions in the 148-gene analyses. (*b*) The results of an AU test comparing the ML tree and four trees that represent alternative positions of *Barthelona* sp. strain PAP020. (Online version in colour.)

pathways in the *Trichomonas* hydrogenosome are schematically shown in electronic supplementary material, figure S4). Thus, we surveyed the transcriptomic data from strain PAP020 for transcripts encoding hydrogenosomal/MRO proteins that are homologous to *Trichomonas* proteins localized in the hydrogenosome. We additionally searched for the MRO proteins in strain PAP020 by using yeast mitochondrial proteins as the queries, albeit no additional candidate was detected. Strain PAP020 was predicted to possess the MRO protein candidates involved in hydrogen production, pyruvate metabolism, amino acid metabolism, Fe–S cluster assembly, the antioxidant system and protein modification (electronic supplementary material, table S2). In figure 4*b*, we mapped the results from the survey on the above-mentioned pathways in the *Trichomonas* hydrogenosome—purple and grey ellipses represent the proteins found and not found, respectively. Purple ellipses with borders represent the MRO protein candidates predicted to have a mitochondrial targeting signal (MTS). Although the overall function of the MRO of strain PAP020 is similar to that of the *Trichomonas* hydrogenosome, we failed to identify some of the key MRO proteins (figure 4*b*). For instance, chaperonin 60/10 (Cpn60/10), mitochondrial-processing peptidase (MPPα/β), hydrogenase maturase (HydE/F/G) and both of

the two enzymes for anaerobic ATP generation through substrate-level phosphorylation [acetate : succinate CoA transferase (ASCT) and succinyl-CoA synthase (SCS)] are missing (figure 4*b*). Likewise, the PAP020 data provided no positive support for the presence of the MRO-localized version of malic enzyme (ME) or pyruvate : ferredoxin oxideoreductase (PFO) ('2' in figure 4*b*; see below for the details). It is too naive to accept the repertoire of MRO proteins predicted from the transcriptome data at face value. At the same time, it is highly unlikely that all of the proteins mentioned above are in fact present in strain PAP020 but escaped detection in our survey, as the quality of the transcriptome data generated in this study is at least comparable to that of other fornicates, such as *C. membranifera*, *A. paluster*, *K. bialata* and *D. brevis* (see the BUSCO scores shown in electronic supplementary material, table S3). Indeed, our analysis identified the nucleotide exchange factor for Hsp70 (GrpE), which was found in neither *K. bialata* nor *D. brevis* (electronic supplementary material, table S4).

The MROs in diplomonads are believed to lack substrate-level phosphorylation, as neither ASCT (which transfers coenzyme A from acetyl-CoA to succinate) nor SCS (which phosphorylates ADP to produce ATP coupled with converting

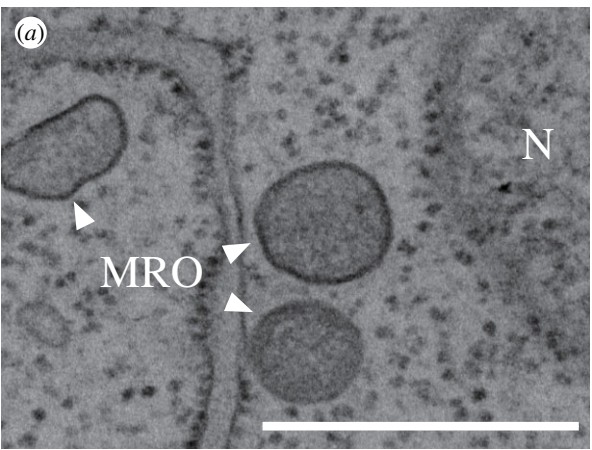

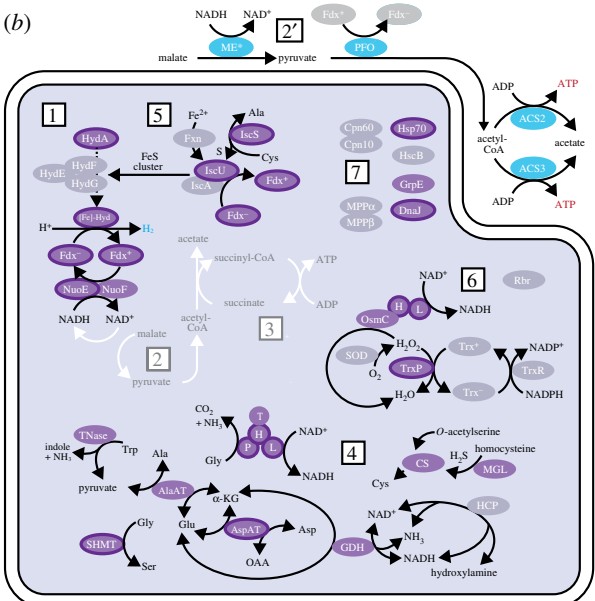

**Figure 4.** Predicted function of the MRO of *Barthelona* sp. strain PAP020. (*a*) Transmission electron micrograph image of MRO of strain PAP020. MROs are shown by arrowheads. The nucleus is labelled as '*N*.' Scale bar, 500 nm. (*b*) MRO protein candidates in strain PAP020. The candidates were mapped on $H_2$-synthesis (labelled as '1'), pyruvate metabolism (2), substrate-level phosphorylation (3), amino acid metabolism (4), Fe–S cluster assembly (5), the antioxidant system (6) and protein modification (7) in the *Trichomonas* hydrogenosome (see electronic supplementary material, figure S4). As we predict that neither pyruvate metabolism nor substrate-level phosphorylation is present in the MRO, the two pathways are shown in white arrows. For the same reason, the MRO enzymes involved in pyruvate metabolism and substrate-level phosphorylation are omitted. Purple ellipses (dark grey ellipses in the grey-scale figure) indicate that the transcripts encoding hydrogenosomal/MRO protein candidates were detected in the PAP020 RNA-seq data. In the case of the ellipses surrounded by borders, their N-termini were predicted as transit peptides for mitochondria/MRO by MitoFates [41] and/or NommPred [42]. The purple/dark grey ellipses with no border indicate putative hydrogenosomal/MRO protein candidates lacking N-terminal sequence information or those with N-terminal extensions that were not predicted as mitochondria/MRO localizing by MitoFates or NommPred. For the proteins represented by grey ellipses (light grey ellipses in the grey-scale figure), we detected no corresponding transcript in the RNA-seq data. Two MEs found in this study were fused with MDH (shown as 'ME*'). In this figure, we assume that pyruvate metabolism occurs in the cytosol of strain PAP020 (labelled as '2''). Nevertheless, the MRO localizations of both MDH–ME and PFO cannot be discarded (see Discussion). See electronic supplementary material, table S2 for the complete protein names. Ala, alanine; Asp, aspartic acid; Cys, cysteine; Glu, glutamic acid; Gly, glycine; α-KG, α-ketoglutaric acid; Trp, tryptophan; OAA, oxaloacetic acid. (Online version in colour.)

succinyl-CoA back to succinate) was found in the genome data of two extensively studied diplomonads, *Giardia intestinalis* and *Spironucleus salmonicida*. Instead, diplomonads are known to generate ATP by acetyl-CoA synthase in the cytosol (ACS1). Similarly, neither ASCT nor SCS was found in the transcriptome data of *D. brevis*, the closest relative of diplomonads (while ACS1 was detected), Leger *et al.* [29] proposed that anaerobic ATP synthesis was lost prior to the separation of *D. brevis* and diplomonads. Significantly, strain PAP020 appeared to be similar to *D. brevis* and *G. intestinalis* in terms of the presence/absence of the enzymes involved in ATP synthesis. Our survey of the transcriptome data from strain PAP020 failed to detect ASCT or SCS, but did identify two distinct acetyl-CoA synthases, both of which are likely localized in the cytosol (see below for the details). These findings prompt us to propose that strain PAP020 has lost anaerobic ATP synthesis in the MRO but generates ATP in the cytosol similar to diplomonads and *D. brevis*.

We here designated two distinct ACS sequences in strain PAP020 as ACS2 and ACS3. Although the transcripts encoding both ACS versions most likely cover their N-termini, neither of them was predicted to bear the typical signal to be localized in mitochondria or MROs (i.e. an inferred N-terminal transit peptide). The abundances of the ACS2 and ACS3 transcripts in strain PAP020 were 717.63 and 626.81 TPM [43], respectively, implying that the two *Barthelona* ACS genes are indistinguishable at the transcription level. We subjected the two ACS sequences to a phylogenetic analysis along with the homologues sampled from diverse bacteria, archaea and eukaryotes (electronic supplementary material, figure S5). The PAP020 ACS2 sequence formed a clade with fornicate 'ACS2' sequences, which Leger *et al.* [29] proposed to be cytosolic enzymes. Thus, we suggest that ACS2 is most likely a cytosolic enzyme in strain PAP020 as well. The ACS phylogeny recovered no strong affinity between PAP020 ACS3 sequence and other homologues (electronic supplementary material, figure S5). Neither of our analyses on the ACS3 sequence provided any positive support for MRO localization, and we tentatively consider ACS3 as a cytosolic enzyme in strain PAP020. Altogether, we propose that strain PAP020 depends entirely on ATP in the cytosol, including by the two cytosol-localizing ACS.

We here propose that strain PAP020 retains pyruvate metabolism in the cytosol, not in the MRO (figure 4*b*). Both of the two PAP020 ME sequences appeared to be proceeded by malate dehydrogenase (MDH) sequences, one of which showed an apparent similarity to the cytosolic homologues and the other to the bacterial homologues (shown as 'ME*'). Further, no MTS was predicted in either of the two MDH–ME fusion proteins (electronic supplementary material, figure S6A). The analyses of the PFO sequences found in strain PAP020 provide little support for their MRO localization. The N-termini of two out of the three PFO sequences are incomplete and their subcellular localizations remain uncertain, while no MTS was predicted for the PFO sequence with the complete N-terminus (electronic supplementary material, figure S6B). Altogether, we favour the cytosolic localization of the two enzymes involved in pyruvate metabolism in strain PAP020 over their localization to the MRO (labels 2' and 2, respectively, in figure 4*b*). The former possibility may not be totally unexpected, as both enzymes were experimentally shown to be localized in the cytosol in *G. intestinalis* [54–56]. To pursue the unsettled issues regarding ATP synthesis and pyruvate metabolism in

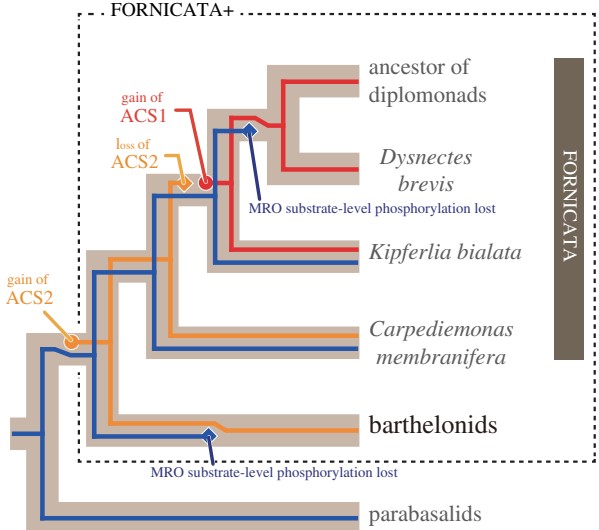

**Figure 5.** Evolution of ATP generation in barthelonids, parabasalids and selected fornicates. In the clade of fornicates and barthelonids (Fornicata+ clade), substrate-level phosphorylation (blue) was lost on two separate branches. The cytosolic ACS2 (yellow), which was established at the base of the Fornicata + clade, was replaced by an evolutionarily distinct type of ACS (ACS1; red) during the evolution of fornicates.

strain PAP020, the precise subcellular localization of ACS2, ACS3, as well as MDH–ME and PFO, need to be confirmed experimentally in the future.

Leger *et al.* [29] proposed a complex evolutionary history for ATP-generating mechanisms in the Fornicata clade, as follows. (i) The ancestral fornicate species possessed both substrate-level phosphorylation in the MRO as well as ACS2 in the cytosol. (ii) Substrate-level phosphorylation has been inherited vertically by the extant fornicate species, except *D. brevis* and diplomonads (see below). (iii) During the evolution of Fornicata, the ancestral cytosol-localizing ACS (i.e. ACS2) was replaced by an evolutionarily distinct ACS (ACS1). (iv) The redundancy in the ATP-generating system allowed the secondary loss of substrate-level phosphorylation in the MRO prior to the separation of the *D. brevis* plus diplomonad clade. We here extend the scenario proposed by Leger *et al.* [29] by incorporating the data from *Barthelona* sp. strain PAP020 (figure 5). Acquisition of ACS2 was hypothesized at the base of the Fornicata clade [29],

but after assessing the data from stain PAP020, this particular event needs to be pushed back at least to the common ancestor of fornicates and barthelonids, as strain PAP020 and multiple early branching CLOs (e.g. *C. membranifera*) share ACS2. It is noteworthy that acquisition of ACS2 may extend back to the last common metamonad ancestor, since a possibly directly related ACS2 is also present in *Paratrimastix* (electronic supplementary material, figure S5). Secondly, as barthelonids are distantly related to *D. brevis* and diplomonads, loss of substrate-level phosphorylation in barthelonid MROs, if this is the case, can be assumed to have occurred independently from the loss in the common ancestor of *D. brevis* and diplomonads (highlighted by blue diamonds in figure 5). Further, barthelonids and the common ancestor of *D. brevis* and diplomonads seem to have accommodated the loss of MRO-localized substrate-level phosphorylation via possessing evolutionarily distinct ACS homologues (ACS2 and ACS1, represented by yellow and red lines, respectively, in figure 5). Finally, pyruvate metabolism might have been relocated from the MRO to the cytosol in strain PAP020 as seen in *G. intestinalis* [54–56].

**Ethics.** Permits for collecting marine resources were obtained from Ministry of Natural Resources, Environment, and Tourism, Palau (permit no. RE-11-18 at 2011, RE17-27 at 2017).

**Data accessibility.** The SSU rDNA sequences of *Barthelona* spp. were deposited in DDBJ database under the accession nos. LC506386–LC506390. The transcriptome data of strain PAP020 were deposited in DDBJ Sequence Archive under the accession nos. DRA009139 and DRA009140. The assembled transcriptomes of strain PAP020 and phylogenetic alignments analysed in this study are available from the Dryad Digital Repository: https://doi.org/10.5061/dryad.3tx95x6bn [57].

**Competing interests.** We declare we have no competing interests.

**Funding.** This work was supported in part by a fund from the grants from the Japan Society for the Promotion of Science (grant nos. 18KK0203 and 19H03280 awarded to Y.I.; grant nos. 15H05231 and 19KK0185 to T.H.) and by the 'Tree of Life' research project of University of Tsukuba, as well as the Natural Sciences and Engineering Research Council of Canada (grant no. 298366-2014 to A.G.B.S.).

**Acknowledgements.** We thank Dr Noèlia Carrasco for providing access to the Delta de l'Ebre sampling location for strain LRM2. The phylogenetic analyses conducted in this work have been carried out under the 'Interdisciplinary Computational Science Program' in the Center for Computational Sciences, University of Tsukuba. The calculation of transcript levels of MRO gene candidates has been carried out on the NIG supercomputer ROIS National Institute of Genetics.

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
