## [Reviewer comments · Proceedings of the Royal Society B: Biological Sciences]

Review History

RSPB-2019-2633.R0 (Original submission)

Review form: Reviewer 1

Recommendation

Major revision is needed (please make suggestions in comments)

Scientific importance: Is the manuscript an original and important contribution to its field?

Excellent

General interest: Is the paper of sufficient general interest?

Good

Quality of the paper: Is the overall quality of the paper suitable?

Good

Is the length of the paper justified?

Yes

Should the paper be seen by a specialist statistical reviewer?

No

Do you have any concerns about statistical analyses in this paper? If so, please specify them explicitly in your report.

No

It is a condition of publication that authors make their supporting data, code and materials available - either as supplementary material or hosted in an external repository. Please rate, if applicable, the supporting data on the following criteria.

Is it accessible?

Yes

Is it clear?

Yes

Is it adequate?

Yes

Do you have any ethical concerns with this paper?

No

Comments to the Author

Yazaki et al. cultured a group of "orphan" eukaryotes, and performed comparative genomic and phylogenomic analyses that place this lineage deep within Fornicates (Metamonada). The Metamonads are a very under-explored group and analyses of novel diversity are very valuable. This paper is also valuable in that it combines protist isolation and culturing with omics analyses, a very worthwhile approach. Combined with interesting inferences about early metamonad and fornicate evolution, this is a very nice manuscript. However, I have some criticisms about analysis of the transcriptome data, and therefore the associated inferences, and I think that addressing these would substantially improve the manuscript.

1. The figures are good, but are there associated legends somewhere? I could not find any in the PDF.
2. MRO proteins in Barthelona were inferred with searches that used proteins from Giardia and Trichomonas MROs as queries. It seems that this approach might miss other MRO-targeted proteins (e.g. ancestral mitochondrial proteins) that have been lost from Trichomonas or Giardia. From that perspective, it is perhaps not surprising that the MRO appears to be very reduced. I think that it would be worth searching for other possible mitochondrial proteins in the transcriptome dataset. One way to do this would be to search the transcripts for sequences with better hits to mitochondrial proteins in other eukaryotes than to a control dataset of potential bacterial contaminants. Alternatively, trees could be inferred for each gene in the transcriptome, and the subset of genes that are monophyletic in eukaryotes could be investigated for mitochondrial localisation/function or, more broadly, their metabolic roles. This same limitation applies to the other inferences about Barthelonid metabolism: the approach taken was a very targeted one, based on genes known to function in other Metamonads. But the transcriptome data generated is a fantastic resource, and could be mined more thoroughly than has been done here.
3. The authors used site-stripping to investigate phylogenetic position of Barthelonids. What about gene-stripping: metamonad genes often have very long branches compared to other eukaryotes; were there any genes for which the metamonad branch lengths were shorter, and what did these say?

4. Some of the relationships in Figures 2 and 3 are heterodox, e.g. various other groups within Archaeplastida (Centrohelid and Cryptista in Figure 3); in Fig. 2 (rDNA tree), Amoebozoa are polyphyletic. I realise that these relationships are not the focus of the present study, and the phylogenomics appear to have been performed carefully. It would nevertheless be worth commenting on why the authors think they obtained this result, perhaps with reference to the relevant support values. It might also be worth commenting on the value of rDNA as a marker for eukaryotic phylogenetics.

Review form: Reviewer 2 (Vladimir Hampl)

Recommendation

Accept with minor revision (please list in comments)

Scientific importance: Is the manuscript an original and important contribution to its field?

Good

General interest: Is the paper of sufficient general interest?

Acceptable

Quality of the paper: Is the overall quality of the paper suitable?

Good

Is the length of the paper justified?

Yes

Should the paper be seen by a specialist statistical reviewer?

No

Do you have any concerns about statistical analyses in this paper? If so, please specify them explicitly in your report.

No

It is a condition of publication that authors make their supporting data, code and materials available - either as supplementary material or hosted in an external repository. Please rate, if applicable, the supporting data on the following criteria.

Is it accessible?

N/A

Is it clear?

Yes

Is it adequate?

Yes

Do you have any ethical concerns with this paper?

No

Comments to the Author

The paper describes phylogenetic home for an orphan protist lineage *Barthelona* spp., which is surprisingly located within microaerophilic Metamonada on the basis of the Fornicata clade. The results of phylogenomic analyses are convincing and corroborated by statistical tests. Authors are well-aware of the fact that the classification of barthelonids within Metamonada must await until

Fornicata ultrastructural features are demonstrated by TEM. Absence of TEM is by the way a regrettable feature of this manuscript as it would nicely complement the study, however, I understand that not always all the data are easy to get.

As expected for metamonads, barthelonids harbour mitochondrion-related organelles, which authors metabolically characterised using the transcriptomic data. From the metabolic predictions they concluded that this particular derivative of mitochondrion metabolises pyruvate but does not produce any ATP. If true, it would be a very interesting and unprecedented case of MRO.

The paper is written in a good language, is concise and despite the large amount of information it is pleasantly short.

Major comments:

1. My major concern relates to the presented metabolic map of the organelle suggesting pyruvate oxidation and at the same time no ATP production. I hesitate to believe in this particular aspect of the metabolic scheme until they provide more data on localisation of the key enzymes PFO, ME and ACS to organelle and cytosol respectively. At present, they infer the localisation of ACS from the absence of targeting sequence and relationship to cytosolic ACSs. Quite inconsistently they infer the PFO and ME to be localised in the organelle, although they do not predict targeting sequences. Strictly speaking all these enzymes have been reported to be cytosolic in some species and organellar in others and predictions of targeting in anaerobic protists are known to be non-reliable, so these arguments are weak. Importantly, they do not comment at all the fate of acetyl-CoA produced in their scheme of the organelle. Obviously, this compound cannot accumulate there, but be either metabolised or transported out. There are several enzymes that can metabolise acetyl-CoA besides ACS and ASCT considered by them, for example Alcohol dehydrogenase E, and AcCoA transporters are also known. Authors should investigate the possibilities more thoroughly and not leave this improbable feature of the metabolic map without comment.

2. The absence of Cpn60, Cpn10 and Mpp from the transcriptomic data is clearly due to incompleteness. There are no known MROs without these proteins. This raises the issue of how complete their dataset is, which is relevant to the previous point. They should provide some completeness statistics (like BUSCO) to cover their back. On a related note. I do not understand the reason, why they generated two transcriptomes and then used one for phylogenomics and the other for metabolic reconstruction. What would be the disadvantage of merging the data to have more complete data set and use it for both purposes?

Minor points:

I could not find the figure legends. Probably technical problem on my side. Therefore, I am not certain, what do the thick margins of the ovals in the metabolic scheme mean. I suppose they mark predicted targeting. This probably applies also to the dark pink colour in the table S2, but here some proteins are not predicted but still in dark pink (one HSP70, two GCSP...). They should somewhere clearly state what the code means and make it consistent. I also miss explanation to the numbers in squares; probably in figure legends.

They claim that they were searching for the MRO proteins using queries from *T. vaginalis* and *G. intestinalis*. How could they find complete GCS and SHMT that are not present in neither of the two protists? Generally, wider set of queries would be useful, because with this narrow they may overlook some interesting features.

The results of the phylogenetic tests should be presented in a supplementary table.

“Metamonad” in the title and abstract should not be capitalised.

Lines 82 and 83, space before bracket with citation is missing.

Figure 2, typos in *Rhodomonas salina*, *Drosophila melanogaster* and *Malawimonas jakobiformis*

Decision letter (RSPB-2019-2633.R0)

07-Jan-2020

Dear Dr Yazaki:

I am writing to inform you that your manuscript RSPB-2019-2633 entitled "Barthelonids represent a deep-branching Metamonad clade with mitochondrion-related organelles generating no ATP." has, in its current form, been rejected for publication in Proceedings B.

This action has been taken on the advice of referees, who have recommended that substantial revisions are necessary. With this in mind we would be happy to consider a resubmission, provided the comments of the referees are fully addressed. However please note that this is not a provisional acceptance.

Sincerely,
Professor Gary Carvalho
<mailto:proceedingsb@royalsociety.org>

Associate Editor
Board Member: 1
Comments to Author:

Two experts in the field have reviewed your manuscript. While they agree on the relevance of the work and its general interest, both have identified major methodological problems (e.g. sequences used as seeds in BLAST) and some minor issue (e.g. the lack of figure captions). I think that by implementing the reviewers' comments, this will improve the quality of the work.

Reviewer(s)' Comments to Author:

Referee: 1

Comments to the Author(s)

Yazaki et al. cultured a group of "orphan" eukaryotes, and performed comparative genomic and phylogenomic analyses that place this lineage deep within Fornicates (Metamonada). The Metamonads are a very under-explored group and analyses of novel diversity are very valuable. This paper is also valuable in that it combines protist isolation and culturing with omics analyses, a very worthwhile approach. Combined with interesting inferences about early metamonad and fornicate evolution, this is a very nice manuscript. However, I have some criticisms about analysis of the transcriptome data, and therefore the associated inferences, and I think that addressing these would substantially improve the manuscript.

1. The figures are good, but are there associated legends somewhere? I could not find any in the PDF.
2. MRO proteins in *Barthelona* were inferred with searches that used proteins from *Giardia* and *Trichomonas* MROs as queries. It seems that this approach might miss other MRO-targeted proteins (e.g. ancestral mitochondrial proteins) that have been lost from *Trichomonas* or *Giardia*. From that perspective, it is perhaps not surprising that the MRO appears to be very reduced. I think that it would be worth searching for other possible mitochondrial proteins in the transcriptome dataset. One way to do this would be to search the transcripts for sequences with better hits to mitochondrial proteins in other eukaryotes than to a control dataset of potential bacterial contaminants. Alternatively, trees could be inferred for each gene in the transcriptome, and the subset of genes that are monophyletic in eukaryotes could be investigated for mitochondrial localisation/function or, more broadly, their metabolic roles. This same limitation applies to the other inferences about *Barthelonid* metabolism: the approach taken was a very targeted one, based on genes known to function in other Metamonads. But the transcriptome data generated is a fantastic resource, and could be mined more thoroughly than has been done here.
3. The authors used site-stripping to investigate phylogenetic position of *Barthelonids*. What about gene-stripping: metamonad genes often have very long branches compared to other eukaryotes; were there any genes for which the metamonad branch lengths were shorter, and what did these say?
4. Some of the relationships in Figures 2 and 3 are heterodox, e.g. various other groups within Archaeplastida (*Centrohelid* and *Cryptista* in Figure 3); in Fig. 2 (rDNA tree), Amoebozoa are polyphyletic. I realise that these relationships are not the focus of the present study, and the phylogenomics appear to have been performed carefully. It would nevertheless be worth commenting on why the authors think they obtained this result, perhaps with reference to the relevant support values. It might also be worth commenting on the value of rDNA as a marker for eukaryotic phylogenetics.

Referee: 2

Comments to the Author(s)

The paper describes phylogenetic home for an orphan protist lineage *Barthelona* spp., which is surprisingly located within microaerophilic Metamonada on the basis of the Fornicata clade. The results of phylogenomic analyses are convincing and corroborated by statistical tests. Authors are well-aware of the fact that the classification of *barthelonids* within Metamonada must await until Fornicata ultrastructural features are demonstrated by TEM. Absence of TEM is by the way a regrettable feature of this manuscript as it would nicely complement the study, however, I understand that not always all the data are easy to get.

As expected for metamonads, barthelonids harbour mitochondrion-related organelles, which authors metabolically characterised using the transcriptomic data. From the metabolic predictions they concluded that this particular derivate of mitochondrion metabolises pyruvate but does not produce any ATP. If true, it would be a very interesting and unprecedented case of MRO.

The paper is written in a good language, is concise and despite the large amount of information it is pleasantly short.

Major comments:

1. My major concern relates to the presented metabolic map of the organelle suggesting pyruvate oxidation and at the same time no ATP production. I hesitate to believe in this particular aspect of the metabolic scheme until they provide more data on localisation of the key enzymes PFO, ME and ACS to organelle and cytosol respectively. At present, they infer the localisation of ACS from the absence of targeting sequence and relationship to cytosolic ACSs. Quite inconsistently they infer the PFO and ME to be localised in the organelle, although they do not predict targeting sequences. Strictly speaking all these enzymes have been reported to be cytosolic in some species and organellar in others and predictions of targeting in anaerobic protists are known to be non-reliable, so these arguments are weak. Importantly, they do not comment at all the fate of acetyl-CoA produced in their scheme of the organelle. Obviously, this compound cannot accumulate there, but be either metabolised or transported out. There are several enzymes that can metabolise acetyl-CoA besides ACS and ASCT considered by them, for example Alcohol dehydrogenase E, and AcCoA transporters are also known. Authors should investigate the possibilities more thoroughly and not leave this improbable feature of the metabolic map without comment.

2. The absence of Cpn60, Cpn10 and Mpp from the transcriptomic data is clearly due to incompleteness. There are no known MROs without these proteins. This raises the issue of how complete their dataset is, which is relevant to the previous point. They should provide some completeness statistics (like BUSCO) to cover their back. On a related note. I do not understand the reason, why they generated two transcriptomes and then used one for phylogenomics and the other for metabolic reconstruction. What would be the disadvantage of merging the data to have more complete data set and use it for both purposes?

Minor points:

I could not find the figure legends. Probably technical problem on my side. Therefore, I am not certain, what do the thick margins of the ovals in the metabolic scheme mean. I suppose they mark predicted targeting. This probably applies also to the dark pink colour in the table S2, but here some proteins are not predicted but still in dark pink (one HSP70, two GCSP...). They should somewhere clearly state what the code means and make it consistent. I also miss explanation to the numbers in squares; probably in figure legends.

They claim that they were searching for the MRO proteins using queries from *T. vaginalis* and *G. intestinalis*. How could they find complete GCS and SHMT that are not present in neither of the two protists? Generally, wider set of queries would be useful, because with this narrow they may overlook some interesting features.

The results of the phylogenetic tests should be presented in a supplementary table.

“Metamonad” in the title and abstract should not be capitalised.

Lines 82 and 83, space before bracket with citation is missing.

Figure 2, typos in *Rhodomonas salina*, *Drosophila melanogaster* and *Malawimonas jakobiformis*

Author's Response to Decision Letter for (RSPB-2019-2633.R0)

See Appendix A.

RSPB-2020-1538.R0

Review form: Reviewer 3

Recommendation

Accept with minor revision (please list in comments)

Scientific importance: Is the manuscript an original and important contribution to its field?

Excellent

General interest: Is the paper of sufficient general interest?

Good

Quality of the paper: Is the overall quality of the paper suitable?

Excellent

Is the length of the paper justified?

Yes

Should the paper be seen by a specialist statistical reviewer?

No

Do you have any concerns about statistical analyses in this paper? If so, please specify them explicitly in your report.

No

It is a condition of publication that authors make their supporting data, code and materials available - either as supplementary material or hosted in an external repository. Please rate, if applicable, the supporting data on the following criteria.

Is it accessible?

N/A

Is it clear?

Yes

Is it adequate?

Yes

Do you have any ethical concerns with this paper?

No

Comments to the Author

See uploaded file (See Appendix B).

Review form: Reviewer 4

Recommendation

Accept with minor revision (please list in comments)

Scientific importance: Is the manuscript an original and important contribution to its field?

Good

General interest: Is the paper of sufficient general interest?

Good

Quality of the paper: Is the overall quality of the paper suitable?

Good

Is the length of the paper justified?

Yes

Should the paper be seen by a specialist statistical reviewer?

No

Do you have any concerns about statistical analyses in this paper? If so, please specify them explicitly in your report.

No

It is a condition of publication that authors make their supporting data, code and materials available - either as supplementary material or hosted in an external repository. Please rate, if applicable, the supporting data on the following criteria.

Is it accessible?

Yes

Is it clear?

Yes

Is it adequate?

Yes

Do you have any ethical concerns with this paper?

No

Comments to the Author

I had a chance to read revised resubmitted manuscript by Yazaki et al on the phylogenetic characterization of barthelonids with a special focus on the nature of its (their) mitochondria (or better MRO). As indicated by previous referees I too find the story very interesting, although with many gaps. However, I entirely understand that it may take considerable time to provide biological data on the actual nature of the MRO. Hence, I believe that the manuscript should be published in its current form with several minor changes.

- 1) The title should be toned down. Instead of saying " with mitochondrion-related organelles generating no ATP" to something like "with mitochondrion-related organelles predicted to generate no ATP". It is fair to make clear line between biochemistry and bioinformatics.
- 2) I too find confusing the information about two different transcriptomes generated and analysed in the story. I understand that there is a timeline in the actual work done in the lab but it

is entirely redundant to keep this information in the manuscript. Authors can define two datasets in the methods but it is not necessary to do it in the results section

3) The possible disconnection of the pyruvate oxidation and ATP formation is very exciting but also bit “upsetting” result. Given the obvious limits of the prediction softwares to handle diverged protein sequences I would strongly recommend to show, in the main figure, at least short alignment of the N-terminal regions of both enzymes (PFO and ME) with the orthologues of both the cytosolic (e.g. *Giardia*) and MRO (e.g. *Trichomonas*) localization. This may allow the reader to directly consider both options of enzymes localization.

4) I would also strongly recommend to show the TEM image of the organelle as part of the main figure (perhaps 4) and not as a supplementary data. It is a nice piece of data.

Decision letter (RSPB-2020-1538.R0)

21-Jul-2020

Dear Dr Yazaki:

Your manuscript has now been peer reviewed and the reviews have been assessed by an Associate Editor. The reviewers’ comments (not including confidential comments to the Editor) and the comments from the Associate Editor are included at the end of this email for your reference. As you will see, the reviewers and the Editors have raised some concerns with your manuscript and we would like to invite you to revise your manuscript to address them.

Thank you for the constructive response to referee comments, and your thorough revision. It is pleasing to see that in principle, both of the referees are happy with the revisions, and are keen to encourage publication of your manuscript. However, as you will see, there does remain several issues, that require your careful attention. While individually, these are mainly minor, collectively, it is important that you address each in turn, with a response and justification, in the usual uploaded response letter. There is no need for the manuscript to be seen by external referees again, though I will need to check carefully the additional revisions submitted. You will see issues ranging from the title, through to several aspects relating to clarification that may impede appropriate interpretation as well as elements of your rationale.

When submitting your revision please upload a file under "Response to Referees" in the "File Upload" section. This should document, point by point, how you have responded to the reviewers’ and Editors’ comments, and the adjustments you have made to the manuscript. We require a copy of the manuscript with revisions made since the previous version marked as ‘tracked changes’ to be included in the ‘response to referees’ document.

Research ethics:

Use of animals and field studies:

It is a condition of publication that you make available the data and research materials supporting the results in the article (<https://royalsociety.org/journals/authors/author-guidelines/#data>). Datasets should be deposited in an appropriate publicly available repository and details of the associated accession number, link or DOI to the datasets must be included in the Data Accessibility section of the article (<https://royalsociety.org/journals/ethics-policies/data-sharing-mining/>). Reference(s) to datasets should also be included in the reference list of the article with DOIs (where available).

Please submit a copy of your revised paper within three weeks. If we do not hear from you within this time your manuscript will be rejected. If you are unable to meet this deadline please let us know as soon as possible, as we may be able to grant a short extension.

Best wishes,
Professor Gary Carvalho
mailto: proceedingsb@royalsociety.org

Associate Editor Board Member
Comments to Author:
See Editor comments above.

Reviewer(s)' Comments to Author:
Referee: 3
Comments to the Author(s).
See uploaded file.

Referee: 4
Comments to the Author(s).

I had a chance to read revised resubmitted manuscript by Yazaki et al on the phylogenetic characterization of barthelonids with a special focus on the nature of its (their) mitochondria (or better MRO). As indicated by previous referees I too find the story very interesting, although with many gaps. However, I entirely understand that it may take considerable time to provide biological data on the actual nature of the MRO. Hence, I believe that the manuscript should be published in its current form with several minor changes.

- 1) The title should be toned down. Instead of saying “ with mitochondrion-related organelles generating no ATP” to something like “with mitochondrion-related organelles predicted to generate no ATP”. It is fair to make clear line between biochemistry and bioinformatics.
- 2) I too find confusing the information about two different transcriptomes generated and analysed in the story. I understand that there is a timeline in the actual work done in the lab but it is entirely redundant to keep this information in the manuscript. Authors can define two datasets in the methods but it is not necessary to do it in the results section
- 3) The possible disconnection of the pyruvate oxidation and ATP formation is very exciting but also bit “upsetting” result. Given the obvious limits of the prediction softwares to handle diverged protein sequences I would strongly recommend to show, in the main figure, at least short alignment of the N-terminal regions of both enzymes (PFO and ME) with the orthologues of both the cytosolic (e.g.Giardia) and MRO (e.g.Trichomonas) localization. This may allow the reader to directly consider both options of enzymes localization.
- 4) I would also strongly recommend to show the TEM image of the organelle as part of the main figure (perhaps 4) and not as a supplementary data. It is a nice piece of data.

Author's Response to Decision Letter for (RSPB-2020-1538.R0)

See Appendix C.

Decision letter (RSPB-2020-1538.R1)

11-Aug-2020

Dear Dr Yazaki

I am pleased to inform you that your manuscript entitled "Barthelonids represent a deep-branching metamonad clade with mitochondrion-related organelles predicted to generate no ATP." has been accepted for publication in Proceedings B.

Open Access

Paper charges

Sincerely,

Professor Gary Carvalho

Associate Editor:

Comments to Author:

The authors were able to address all the reviewers' concerns, and I glad to recommend the manuscript for publication.

Best wishes,

Roberto Feuda

Appendix A

Associate Editor

Board Member: 1

Comments to Author:

Two experts in the field have reviewed your manuscript. While they agree on the relevance of the work and its general interest, both have identified major methodological problems (e.g. sequences used as seeds in BLAST) and some minor issue (e.g. the lack of figure captions). I think that by implementing the reviewers' comments, this will improve the quality of the work.

Reviewer(s)' Comments to Author:

Referee: 1

Comments to the Author(s)

Yazaki et al. cultured a group of "orphan" eukaryotes, and performed comparative genomic and phylogenomic analyses that place this lineage deep within Fornicates (Metamonada). The Metamonads are a very under-explored group and analyses of novel diversity are very valuable. This paper is also valuable in that it combines protist isolation and culturing with omics analyses, a very worthwhile approach. Combined with interesting inferences about early metamonad and fornicate evolution, this is a very nice manuscript. However, I have some criticisms about analysis of the transcriptome data, and therefore the associated inferences, and I think that addressing these would substantially improve the manuscript.

1. The figures are good, but are there associated legends somewhere? I could not find any in the PDF.

Each figure now has the corresponding legend.

2. MRO proteins in *Barthelona* were inferred with searches that used proteins from *Giardia* and *Trichomonas* MROs as queries. It seems that this approach might miss other MRO-targeted proteins (e.g. ancestral mitochondrial proteins) that have been lost from *Trichomonas* or *Giardia*. From that perspective, it is perhaps not surprising that the MRO appears to be very reduced. I think that it would be worth searching for other possible mitochondrial proteins in the transcriptome dataset. One way to do this would be to search the transcripts for sequences with better hits to mitochondrial proteins in other eukaryotes than to a control dataset of potential bacterial contaminants. Alternatively, trees could be inferred for each gene in the transcriptome, and the subset of genes that are monophyletic in eukaryotes could be investigated for mitochondrial localisation/function or, more broadly, their metabolic roles. This same limitation applies to the other inferences about *Barthelonid* metabolism: the approach taken was a very targeted one, based on genes known to function in other Metamonads. But the transcriptome data generated is a fantastic resource, and could be mined more thoroughly than has been done here.

As the reviewer commented above, our methodology for the survey of putative MRO proteins in *Barthelona* strain PAP020 could have missed the proteins that have been lost from the *Giardia* or *Trichomonas* MRO. To assess whether strain PAP020 possesses the MRO proteins that have been found in neither *Giardia* nor *Trichomonas* MRO, we repeated the homology search against the PAP020 transcriptome data by using all of the mitochondrial proteins in budding yeast as queries. Regrettably, no additional candidate, which may localize and function in the barthelonid MRO, was identified. Thus, our prediction on the function of the MRO in strain PAP020 remains the same. According to this modification in our similarity search, we added new sentences to the Materials and Methods (p. 5, lines 160-164) and Results and Discussion (p. 7, lines 248-249).

3. The authors used site-stripping to investigate phylogenetic position of *Barthelonids*. What about gene-stripping: metamonad genes often have very long branches compared to other eukaryotes; were there any genes for which the metamonad branch lengths were shorter, and what did these say?

We read the above comment as the recommendation of additional analyses to assess whether the grouping of PAP020 and metamonads is a long-branch attraction (LBA) artifact. Unfortunately, the strategy suggested by the reviewer was found not to be applicable to our phylogenomics (148-gene) alignment. We noticed that PAP020 and metamonads were long branches in the vast majority of the single-gene trees. If we consider only single genes with short branch metamonad/PAP020 sequences, the resultant multigene alignment will be very small. Therefore, we took a different approach to address the reviewer's inquiry.

We generated phylogenetic alignments comprising 30, 60, 90, and 120 randomly sampled genes, and subjected these alignments to the maximum-likelihood bootstrap analyses (The details of the method are described in p. 5, lines 152-157). Briefly, we found that two conflicting phylogenetic signals, one uniting strain PAP020 and fornicates and the other uniting strain PAP020 and parabasalids, and the former appeared to dominate the latter in the analyses of the alignments comprising 90 and 120 genes. The gene subsampling analyses suggested no gene-specific signal that could have biased the phylogenetic inference from the 148-gene alignment severely. The results from the above-mentioned analyses were described in p. 6, lines 204-211 along with a new set of supplementary figures (Figs. S3A-D).

During the gene subsampling analyses, we noticed that 7 out of 50 alignments generated by random sampling of 30 genes ("rs30-gene alignments") failed to unite PAP020 with fornicates or parabasalids with MLBPs of $\geq 50\%$ (See the data points highlighted by red arrowheads in Fig. S3A). In the ML trees inferred from those alignments, all of or subsets of the metamonads (including PAP020) grouped with apparently non-metamonad taxa, e.g., a foraminifer *Quinqueloculina* sp., with high statistical support (See Fig. S4A-G). It is highly likely that *Quinqueloculina* sp. grouped erroneously with metamonads due to LBA stemmed from the divergent sequence nature and low site-coverage of this taxon. Significantly, the analyses of the alignments of randomly sampled 60-120 genes seemingly overcame the LBA artifact between *Quinqueloculina* sp. and metamonads, but placed PAP020 with metamonads. These observations suggest that the phylogenetic affinity between PAP020 and metamonads, particularly fornicates, is unlikely an LBA artifact. These results are discussed in p. 6, lines 207-p. 7, line 221.

4. Some of the relationships in Figures 2 and 3 are heterodox, e.g. various other groups within Archaeplastida (Centrohelid and Cryptista in Figure 3); in Fig. 2 (rDNA tree), Amoebozoa are polyphyletic. I realise that these relationships are not the focus of the present study, and the phylogenomics appear to have been performed carefully. It would nevertheless be worth commenting on why the authors think they obtained this result, perhaps with reference to the relevant support values. It might also be worth commenting on the value of rDNA as a marker for eukaryotic phylogenetics.

To respond to the above comment, we revised the first paragraph in the Results and Discussion (p. 5, line 172-p. 6, line 180). We believe that SSU rDNA is the gene sampled most widely across the tree of eukaryotes and thus the results from the SSU rDNA phylogenies are significant as the "starting points" to explore the phylogenetic positions of the eukaryotes of interest.

Referee: 2

Comments to the Author(s)

The paper describes phylogenetic home for an orphan protist lineage *Barthelona* spp., which is surprisingly located within microaerophilic Metamonada on the basis of the Fornicata clade. The results of phylogenomic analyses are convincing and corroborated by statistical tests. Authors are well-aware of the fact that the classification of barthelonids within Metamonada must await until Fornicata ultrastructural features are demonstrated by TEM. Absence of TEM is by the way a regrettable feature of this manuscript as it would nicely complement the study, however, I understand that not always all the data are easy to get.

As expected for metamonads, barthelonids harbour mitochondrion-related organelles, which authors metabolically characterised using the transcriptomic data. From the metabolic predictions they concluded

that this particular derivative of mitochondrion metabolises pyruvate but does not produce any ATP. If true, it would be a very interesting and unprecedented case of MRO.

The paper is written in a good language, is concise and despite the large amount of information it is pleasantly short.

Major comments:

1. My major concern relates to the presented metabolic map of the organelle suggesting pyruvate oxidation and at the same time no ATP production. I hesitate to believe in this particular aspect of the metabolic scheme until they provide more data on localisation of the key enzymes PFO, ME and ACS to organelle and cytosol respectively. At present, they infer the localisation of ACS from the absence of targeting sequence and relationship to cytosolic ACSs. Quite inconsistently they infer the PFO and ME to be localised in the organelle, although they do not predict targeting sequences. Strictly speaking all these enzymes have been reported to be cytosolic in some species and organellar in others and predictions of targeting in anaerobic protists are known to be non-reliable, so these arguments are weak. Importantly, they do not comment at all the fate of acetyl-CoA produced in their scheme of the organelle. Obviously, this compound cannot accumulate there, but be either metabolised or transported out. There are several enzymes that can metabolise acetyl-CoA besides ACS and ASCT considered by them, for example Alcohol dehydrogenase E, and AcCoA transporters are also known. Authors should investigate the possibilities more thoroughly and not leave this improbable feature of the metabolic map without comment.

We agree with the inconsistency in our interpretations of the subcellular localization of ME, PFO, and ACS in the submitted manuscript. Unfortunately, all of our predictions on subcellular localization of the proteins of interest depend solely on the in silico analyses of the N-terminal amino acid sequences. Therefore, there is a certain room of uncertainty for mitochondrial localization, regardless of the results from the prediction. In the revised manuscript, we discuss two possibilities for the subcellular localization of ME and PFO (p. 8, lines 293-p. 9, 304; Fig. 4A). One possibility is the MRO localization of ME and PFO as discussed in the submitted manuscript. In the first scenario assuming the MRO-localization of ME and PFO (labeled as “2” in the revised Fig. 4A), the transport of acetyl-CoA to the cytosol is suggested in the main text and figure 4A, as we failed to find any transcripts encoding candidates MRO enzymes that can catalyze acetyl-CoA in the transcriptome data. We now discuss the second scenario in which ME and PFO are localized in the cytosol as confirmed experimentally in *Giardia intestinalis*. This possibility is presented as “2” in the revised Fig. 4A. In the revised manuscript, we expressed our preference for the cytosolic pyruvate metabolism over the other (p. 9, lines 299-300), but avoided to discard the other scenario completely.

We left the localization of the two ACS versions as cytosolic in the revised manuscript, as our analyses of their amino acid sequence did not hint at the MRO localization; no MTS has predicted or no phylogenetic affinity to the previously known, cytosolic ACS sequences were recovered (see the results and discussion in p. 8, line 279-292).

2. The absence of Cpn60, Cpn10 and Mpp from the transcriptomic data is clearly due to incompleteness. There are no known MROs without these proteins. This raises the issue of how complete their dataset is, which is relevant to the previous point. They should provide some completeness statistics (like BUSCO) to cover their back. On a related note. I do not understand the reason, why they generated two transcriptomes and then used one for phylogenomics and the other for metabolic reconstruction. What would be the disadvantage of merging the data to have more complete data set and use it for both purposes?

The two RNA-seq analyses had been done in 2013 and 2017 by using HiSeq 2500 and MiSeq platforms, respectively. The first RNA-seq data contains a large amount of the sequences derived from bacteria in the culture medium, as we had not established the method that can separate bacteria and barthelonid cells yet. In the second analysis, we efficiently eliminated bacteria from barthelonid cells before RNA extraction, and thus the second RNA-seq data were anticipated (and indeed appeared) to contain much less bacterial sequences than the first data. As many of the proteins localized in

mitochondria/hydrogenosome/MRO are of bacterial origin, the chance of annotating contaminated bacterial sequences faulty as the *Barthelona* sequences encoding MRO proteins is much lower in the second RNA-seq data than the first one. On the other hand, the majority of the proteins considered in the phylogenomic alignment is mostly eukaryote-specific, so that the first data was sufficient for the construction of a phylogenomic alignment. The reasons we generated two transcriptome data for strain PAP020 are described briefly in the corresponding section in Materials & Methods (p. 3, lines 95-101).

Although not mentioned in the text, we did merge the contig data generated from the two RNA-seq data together and searched for the transcripts encoding putative MRO proteins, as well as those encoding the 148 proteins considered in the phylogenomic analyses. Nevertheless, no extra information was obtained for the phylogenomic analysis or reconstruction of the metabolic pathways in the MRO.

We failed to detect Cpn60/10, MPP α/β , HydE/F/G, SCS, or ASCT in the transcriptome data from strain PAP020 (p. 7, lines 255-260). The comparison of the BUSCO scores for the transcriptome data among strain PAP020 and four fornicates, and the PAP020 data was found to be compatible with others (Table S3). Nevertheless, it is difficult to judge whether each of the proteins listed above is truly absent in strain PAP020 based solely on the transcriptome data (p. 7, line 249-255). In the revised manuscript, we “proposed” (not “concluded” as we did in the submitted manuscript) the absence of substrate-level phosphorylation in the MRO of strain PAP020 by referring the ATP-generating mechanism in *Dysnectes brevis* and diplomonads (p. 8, line 267-278). Along with this revision, we changed the wording in Abstract (p. 1, line 25), Results & Discussion (p. 9, line 290), and the legend for Fig. 4 (p. 13, lines 501-504).

Minor points:

I could not find the figure legends. Probably technical problem on my side. Therefore, I am not certain, what do the thick margins of the ovals in the metabolic scheme mean. I suppose they mark predicted targeting. This probably applies also to the dark pink colour in the table S2, but here some proteins are not predicted but still in dark pink (one HSP70, two GCSP...). They should somewhere clearly state what the code means and make it consistent. I also miss explanation to the numbers in squares; probably in figure legends.

Each figure now has the corresponding legend.

They claim that they were searching for the MRO proteins using queries from *T. vaginalis* and *G. intestinalis*. How could they find complete GCS and SHMT that are not present in neither of the two protists? Generally, wider set of queries would be useful, because with this narrow they may overlook some interesting features.

We forgot to mention that we use putative MRO proteins found in other fornicates including *Dysnectes brevis* and *Kipferlia bialata* as queries in the submitted manuscript. We revised the corresponding sentence in the revised manuscript (p. 5, lines 162-163). In addition, we repeated our survey of MRO protein candidates by using mitochondrial proteins in yeast to respond to a comment from reviewer 1. Please see the section “Prediction of proteins localized in the mitochondrion-related organelle in *Barthelona* sp. strain PAP020” in the Materials & Methods (p. 5, lines 163-164).

The results of the phylogenetic tests should be presented in a supplementary table.

We presented the result of an approximately unbiased test in the revised Figure 3B.

“Metamonad” in the title and abstract should not be capitalised.

Corrected.

Lines 82 and 83, space before bracket with citation is missing.

Corrected.

Figure 3, typos in *Rhodomonas salina*, *Drosophila melanogaster* and *Malawimonas jakobiformis*

Corrected.

Appendix B

RSPB-2020-1538

Barthelonids represent a deep-branching metamonad clade with mitochondrion-related organelles generating no ATP by Yazaki *et al.*

There has been, and is, considerable interest in the evolutionary events surrounding the origin of eukaryotes. Part of the research focuses on the role of mitochondria in this process. Many of the supposedly 'primitive' eukaryotic clades contain anaerobic species with highly modified mitochondria such as mitosomes and hydrogenosomes. The metamonads received particular interest these unusual organelles were discovered in these groups.

The barthenolids are a group of anaerobic microbial eukaryotes of uncertain affiliation. In addition, the metabolic status of their mitochondria has not been studied either. Yazaki and colleagues have provided detailed information about these two questions. Their work clearly showed that barthenolids are metamonads, and more in particular the Fornicata. As expected, the initial SSU based phylogeny were not resolving the placement of the various studied barthenolids (although robustly placing them together). A subsequent multi-gene concatenated phylogenomic analysis clearly demonstrated the deep-branching metamonad placement of the barthenolids. As barthenolids grow with bacteria as their food, the possibility exists that contaminating bacterial sequences had entered the multi-gene phylogeny. Yazaki *et al.* corrected for this by performing a series of subset concatenated multi-gene phylogenies that principally all resulted in the same phylogeny. Their well-executed and robust phylogenomic analyses leaves little room for alternative interpretations.

In addition, Yazaki *et al.* used the transcriptomic data to assess the putative mitochondrial biochemical pathways. Using *Trichomonas* and *Saccharomyces* hydrogenosomal and mitochondrial pathways as queries, they assessed the presence and completeness of the barthenolid organelle. Their analyses clearly indicate a much reduced mitochondrial biochemistry in line with that found for other closely related diplomonad species.

Overall, this is a well-executed study that sheds some light on a eukaryotic orphan group by firmly placing it in its phylogenetic place in the tree of life and by unravelling its putative mitochondrial metabolic pathway.

I have not any serious issues that need to be addressed but there are few things that I would suggest to change.

In order of appearance:

Line 25/27: 'We here propose that strain PAP020 is incapable of generating ATP in the MRO, as no mitochondrial/MRO enzymes involved in substrate-level phosphorylation were detected'. Yes, but mitochondrial ATP is normally produced by harvesting the electrochemical gradient generated via the electron transport chain and not via substrate-level phosphorylation. That might well be the case for most MROs but that is something non-initiated readers would know. So, this needs some additional clarification by mentioning that MROs in metamonads (or MROs in general perhaps) do produce ATP via substrate-level phosphorylation unlike classic aerobic mitochondria who produce it via the electron transport chain and oxidative phosphorylation. This also links to the apparent lack of checking if there are any components of the electron transport chain present (Complex I-IV and the ATPase). Please indicate clearly in the text that those were checked for but not found. If not, than the authors do need to check for the presence of any component of the electron transport chain!

Line 73: In this English sentence, it has to be the Ebro Delta, not the Ebre Delta. Later, in the acknowledgement, the Catalan Ebre is fine but not in this sentence as the language is different.

Line 102-106: Although the second RNASeq run specifically mentions it, in this first RNASeq experiment, no poly-A enrichment step was used? That would have been good. That the authors extensively try to correct for the possible inclusion of bacterial sequences corrects for the possible lack of that enrichment step.

Line 237: As mentioned before, the lack of substrate-level phosphorylation does need some clarification. Alternatively, a heading that mentions that there is no ATP production in the MRO is a possible solution as well.

Line 257-260: This needs changing. The very likely absence of SCS, ASCT, PFO, and ME from the organelle is made rather confusing by their inclusion in the figure as possibly organellar. I would recommend to remove these enzymes from the MRO pathways in figure 4 to avoid any confusion. There is no evidence they are organellar so there is no need to suggest they are in the figure. If the authors want to keep the option open that they might turn out to be organellar, then they can say so in the text (as they do) but not in the figure. This will avoid others looking at the figure and assuming these steps are in the organelle as the figure suggests.

Line 291-291: Similar, there is no need to invoke organellar localisation so the dotted line in figure 4 can go.

Line 291: 'We suspect whether strain PAP020 retains pyruvate metabolism in the MRO'. This is a confusing sentence and it is not clear what is meant?

Line 302-304: As mentioned above, this does indeed need further study so please remove these steps from the organellar pathways in figure 4.

Line 500-504: And again, please remove or change this as there is no need to invoke organellar localisation in the lack of supporting evidence. In addition, closely related species do not have these steps in their mitochondrial organelle either.

Appendix C

Responses to the comments/suggestions from referee 3

- 1) Line 25/27: ‘We here propose that strain PAP020 is incapable of generating ATP in the MRO, as no mitochondrial/MRO enzymes involved in substrate-level phosphorylation were detected’. Yes, but mitochondrial ATP is normally produced by harvesting the electrochemical gradient generated via the electron transport chain and not via substrate-level phosphorylation. That might well be the case for most MROs but that is something non-initiated readers would know. So, this needs some additional clarification by mentioning that MROs in metamonads (or MROs in general perhaps) do produce ATP via substrate-level phosphorylation unlike classic aerobic mitochondria who produce it via the electron transport chain and oxidative phosphorylation. This also links to the apparent lack of checking if there are any components of the electron transport chain present (Complex I-IV and the ATPase). Please indicate clearly in the text that those were checked for but not found. If not, then the authors do need to check for the presence of any component of the electron transport chain!

The point raised above is critical for the readers who are not familiar with MROs. No component of the electron transport chain (ETC) or ATP synthase was detected by the BLAST against the transcriptome data of strain PAP020 using the *Saccharomyces* orthologues as queries. Thus, strain PAP020 is most likely incapable to generate ATP via the ETC as do aerobic organisms with typical mitochondria. We briefly described the putative absence of the proteins involved in the ETC and ATP synthase in strain PAP020 in p. 7, lines 246-250.

- 2) Line 73: In this English sentence, it has to be the Ebro Delta, not the Ebre Delta. Later, in the acknowledgement, the Catalan Ebre is fine but not in this sentence as the language is different. Corrected. Please see p. 3, line 73,
- 3) Line 102-106: Although the second RNASeq run specifically mentions it, in this first RNASeq experiment, no poly-A enrichment step was used? That would have been good. That the authors extensively try to correct for the possible inclusion of bacterial sequences corrects for the possible lack of that enrichment step.

In the first analysis, a biotech company constructed the cDNA library from poly-A tailed RNAs. We clarified this point by revising a sentence in p. 4, lines 104-105.

- 4) Line 237: As mentioned before, the lack of substrate-level phosphorylation does need some clarification. Alternatively, a heading that mentions that there is no ATP production in the MRO is a possible solution as well.

We changed the title of the corresponding section from “Lack of substrate-level phosphorylation in the mitochondrion-related organelle of *Barthelona* sp. PAP020” to “No ATP production in the mitochondrion-related organelle of *Barthelona* sp. PAP020”. Please see p. 7, line 241.

- 5) Line 257-260: This needs changing. The very likely absence of SCS, ASCT, PFO, and ME from the organelle is made rather confusing by their inclusion in the figure as possibly organellar. I would recommend to remove these enzymes from the MRO pathways in figure 4 to avoid any confusion. There is no evidence they are organellar so there is no need to suggest they are in the figure. If the authors want to keep the option open that they might turn out to be organellar, then they can say so in the text (as they do) but not in the figure. This will avoid others looking at the figure and assuming these steps are in the organelle as the figure suggests.

We revised figure 4A in the submitted manuscript by incorporating the above comment (The revised figure is numbered as 4B in the revised manuscript). As the result, the figure now has no information regarding the enzymes involved in pyruvate metabolism and substrate-level phosphorylation. For the readers' convenience, we provide a new supplementary figure, which displays the metabolic pathway of the *Trichomonas* hydrogenosome (Fig. S4), as the reference of Fig. 4B.

- 6) Line 291-291: Similar, there is no need to invoke organellar localisation so the dotted line in figure 4 can go.

We removed the dotted line from the new figure for the metabolic pathways in the PAP020 MRO (Fig. 4B).

- 7) Line 291: ‘We suspect whether strain PAP020 retains pyruvate metabolism in the MRO’. This is a confusing sentence and it is not clear what is meant?

We reworded the sentence. Please see p. 9, line 300.

- 8) Line 302-304: As mentioned above, this does indeed need further study so please remove these steps from the organellar pathways in figure 4.

Please see the new version of the figure (i.e. Fig. 4B).

- 9) Line 500-504: And again, please remove or change this as there is no need to invoke organellar localisation in the lack of supporting evidence. In addition, closely related species do not have these steps in their mitochondrial organelle either.

We removed the corresponding sentence from the legend of Fig. 4 (p. 13).

Responses to the comments/suggestions from referee 4

- 1) The title should be toned down. Instead of saying “with mitochondrion-related organelles generating no ATP” to something like “with mitochondrion-related organelles predicted to generate no ATP”. It is fair to make clear line between biochemistry and bioinformatics.
According to referee’s suggestion, we revised the title as “...with mitochondrion-related organelles predicted to generate no ATP”. Please see the title page.
- 2) I too find confusing the information about two different transcriptomes generated and analysed in the story. I understand that there is a timeline in the actual work done in the lab but it is entirely redundant to keep this information in the manuscript. Authors can define two datasets in the methods but it is not necessary to do it in the results section
We rechecked both submitted and revised manuscripts not to be repetitive. We confirmed that the two RNA-seq analyses are described only in the Materials & Method section in the revised manuscript (p. 3, line 94-p. 4, line 115).
- 3) The possible disconnection of the pyruvate oxidation and ATP formation is very exciting but also bit “upsetting” result. Given the obvious limits of the prediction softwares to handle diverged protein sequences I would strongly recommend to show, in the main figure, at least short alignment of the N-terminal regions of both enzymes (PFO and ME) with the orthologues of both the cytosolic (e.g. *Giardia*) and MRO (e.g. *Trichomonas*) localization. This may allow the reader to directly consider both options of enzymes localization.
We prepared the alignments of the PFO and ME amino acid sequences of strain PAP020, *Trichomonas*, *Giardia*, and *E. coli* for the readers’ convenience. However, as the referee commented, we cannot conclude the subcellular localizations of the PAP020 ME and PFO based solely on the sequence information—many of them are incomplete and, even if the N-terminus is

completed, no MTS was predicted. Thus, these figures are provided as a supplementary figure (Fig. S6).

- 4) I would also strongly recommend to show the TEM image of the organelle as part of the main figure (perhaps 4) and not as a supplementary data. It is a nice piece of data.

We revised figure 4 to reflect the referee's suggestion. You can find the TEM image as figure 4A in the revised manuscript. The method and legend for Fig. 4A can be found in p. 5, lines 158-170 and p. 13, lines 473-474, respectively. The schematic figure for the evolution of ATP generation in the Fornicata+ clade is provided as Fig. 5 in the revised manuscript.